# A Historical Review of Management Options Used against the Stable Fly (Diptera: Muscidae)

**DOI:** 10.3390/insects11050313

**Published:** 2020-05-15

**Authors:** David Cook

**Affiliations:** Department of Primary Industries and Regional Development, South Perth, WA 6151, Australia; david.cook3@dpird.wa.gov.au; Tel.:+61-8-9368-3084

**Keywords:** insecticide, biopesticide, entomopathogenic fungi, insect growth regulator, wasps, parasitoid, traps, sanitation, repellent, compaction

## Abstract

The stable fly, *Stomoxys calcitrans* (L.) (Diptera: Muscidae), remains a significant economic pest globally in situations where intensive animal production or horticultural production provide a suitable developmental medium. Stable flies have been recorded as pests of livestock and humans since the late 1800s to early 1900s. Over 100 years of research has seen numerous methodologies used to control this fly, in particular to protect cattle from flies to minimise production losses. Reduced milk production in dairy cows and decreased weight gain in beef cattle account for losses in the US alone of > $2000 million annually. Rural lifestyles and recreation are also seriously affected. Progress has been made on many control strategies against stable fly over a range of chemical, biological, physical and cultural options. This paper reviews management options from both a historical and a technical perspective for controlling this pest. These include the use of different classes of insecticides applied to affected animals as toxicants or repellents (livestock and humans), as well as to substrates where stable fly larvae develop. Arthropod predators of stable flies are listed, from which potential biological control agents (e.g., wasps, mites, and beetles) are identified. Biopesticides (e.g., fungi, bacteria and plant-derived products) are also discussed along with Integrated Pest Management (IPM) against stable flies for several animal industries. A review of cultural and physical management options including trapping, trap types and methodologies, farm hygiene, scheduled sanitation, physical barriers to fly emergence, livestock protection and amendments added to animal manures and bedding are covered. This paper presents a comprehensive review of all management options used against stable flies from both a historical and a technical perspective for use by any entomologist, livestock producer or horticulturalist with an interest in reducing the negative impact of this pest fly.

## 1. Introduction

The stable fly, *Stomoxys calcitrans* (Diptera: Muscidae), remains a significant economic pest globally in places where either intensive animal production or horticultural production provide a suitable developmental medium. Accumulations of animal manure, soiled bedding, wet hay and silage [1] and rotting plant material (reject produce, abandoned crops, processing waste) can produce substrates that support stable fly larval development [2,3,4].

Stable flies have been recorded as pests of livestock and humans dating back to the late 1800s to early 1900s. The first record of stable flies affecting livestock was in 1889, when a new cattle pest was reported in the American Naturalist by Williston [5]. Protecting cows from flies (including stable flies) was noted in Connecticut, US in the early 1900s [6]. By the early 1900–1910s, articles were being published on the life history and bionomics of stable flies [6,7,8,9]. Reduced weight gain can occur with as few as 20 flies per animal [10,11]. Cattle often bunch together when attacked by stable flies to avoid being bitten; this often leads to heat stress and reduced feeding [12,13,14]. Serum levels of the stress hormone cortisol increased in dairy cows in response to higher numbers of stable flies on them and their associated fly-dislodging behaviors [15]. Their painful bites can reduce milk production in dairy cows, decrease weight gain in beef cattle and affect feed efficiency [16] with national losses in cattle production for the US alone being over $2 billion [17]. Rural lifestyle and recreation are also seriously affected in peri-urban and rural communities [18,19].

Several factors make stable flies difficult to control: (i) this fly only visits its host briefly to obtain a blood meal, making chemical control difficult; (ii) larval developmental sites are widespread and often ephemeral; and iii) stable flies are strong fliers, hence adequate control requires area-wide efforts [20]. Progress has been made on many control options against this blood-feeding fly covering a broad range of chemical, biological, physical and cultural options along with trapping for monitoring and fly reduction [4]. A previous review of the importance of *S. calcitrans* as a livestock pest in 2018 only briefly discussed chemical, biological and cultural control options [21]. This paper presents a comprehensive review of management options from both a historical perspective and as a resource for any entomologist, livestock producer or horticulturalist with an interest in control and management strategies that have been used previously to reduce the impact of this pest fly.

## 2. Chemicals

### 2.1. Insecticides

The use of insecticides against stable flies dates back to the early 1900s, where the first record in the literature on stable fly control was by Beach and Clark [6], who wrote about protecting cows from flies. Over that century, insecticide use moved from dichloro-diphenyl-trichloroethane (DDT), organochlorines, carbamates and organophosphates to pyrethrins, both on animals directly and in situations where larval infestations of stable fly were found (e.g., rotting hay, soiled animal bedding, rotting vegetable matter) [22]. Various authors have reviewed a range of pesticides against stable flies, with Mount et al. [23,24] providing the most comprehensive evaluations. These two laboratory studies combined tested >230 compounds against adults and 390 compounds against larvae in treated media. Subsequently, the toxicity of various insecticides to stable flies has been reviewed [25,26,27,28].

Since the early 1930s, significant discoveries led to the proliferation of new synthetic pesticides including organochlorines, organophosphates and pyrethroids [29]. Plant-derived pesticides such as rotenone (isoflavone) and pyrethrum were applied as dusts on cattle to control stable flies in 1936 [30]. DDT was one of the first modern synthetic insecticides developed to combat mainly insect-borne human diseases (e.g., malaria and typhus) among both military and civilian populations. DDT was highly effective against stable flies both in animal barns [31,32] and in larval infestations in shore deposits of marine grass [33]. However, within a decade, DDT resistance was detected in stable fly populations [34,35,36]. Dichloro-diphenyl-dichloroethane (DDD) is a metabolite of DDT considered to be less toxic to animals than DDT. Applications of 1% spray solutions of DDD on cattle provided equivalent control of stable flies as DDT [37]. The organochlorine dieldrin was widely used between the 1950s and 1970s in agriculture [38] but, similar to DDT, it came and went within a decade for use against stable flies [39] once resistance quickly developed [40].

Pine tar creosote was shown as early as 1917 to protect dairy cattle from stable flies [41]. Creosote oil and diesel controlled larval infestations of stable flies in marine grass accumulations [42,43]. Similarly, gas condensate produced by the carbonisation of coal was tested as a cheap and readily available toxic agent against stable fly larvae infesting grasses, peanut vine litter and celery crop residues [44]. However, mass accumulations of waste celery could not be adequately treated by insecticides or gas condensate due to their sheer volume (260,000 m^3^) and density of stable fly larvae (≈ 2 million/m^3^) [22]. Preventative spraying schedules to reduce anticipated outbreaks of stable flies were successful for dairy farms in the 1960s using carbamates and organophosphates [45]. Fogging and spraying with various insecticides were effective at reducing stable flies in milking sheds and barns (DDT [31]; dimethoate [46]; pyrethrin [47]; resmethrin [48].

Methoxychlor (organochloride), toxaphene and DDT (now mostly obsolete due to environmental and human health concerns) caused rapid knockdown and death of stable flies and maintained a residual efficacy for several months [49]. This led to organochlorines being applied to the exterior surfaces of animal barns to reduce stable flies. The next major group of chemicals used against stable flies were the organophosphates (e.g., dimethoate, chlorpyifos, crotoxyphos), thiophosphates (e.g., dichlorvos) and phosphothiorates (e.g., coumaphos), which have been available as insect control agents since the 1950s. Following the demise of DDT, organophosphates quickly followed as the pesticides of choice against stable flies. Stable flies were controlled on Norwegian farms with trichlorfon [35] and crotoxyphos was used against stable flies on dairy cattle [50]. Aerial application of insecticides to dairy farms and feedlots resulted in reductions in stable fly numbers between a maximum of 60% on dairy farms and up to 90% on feedlots when assessed 24 h after application of the organophosphates naled, fenthion and dichlorvos [51].

The next major group of pesticides used against stable flies were the synthetic pyrethroids (SP), which were far more effective at killing insects than DDT, but without the negative impacts on human health [52] and the environment [53]. Cyclethrin was used on cattle and effective control of stable flies was observed [54]. The pyrethroid resmethrin applied as a dust by compressed gas into a 425 m^3^ barn effectively killed all flies including stable, house and lesser house fly [48]. Schmidt et al. [55] evaluated an SP on cattle and observed effective control of stable flies for 8 d after treatment. Permethrin applied as a backline treatment to cattle controlled stable flies for up to 2 weeks [56] and improved milk yield in dairy cows by 0.8 kg/animal [57]. Permethrin-soaked tape attached to the ears and tails of dairy cattle provided up to 10 weeks control of stable flies [58,59].

Stable fly populations in Mexican dairy farms were susceptible to permethrin when assessed in 2005 [60]. Five years later, signs of permethrin resistance were evident in Florida stable fly populations [61] and first noted in European populations soon after [62,63]. Most recently, the widespread use of insecticides against stable fly outbreaks in Brazil have resulted in the populations around sugar cane mills in the state of Mato Grosso do Sul becoming resistant to cypermethrin, with resistance factors up to 39 [64]. Similarly, Reissert-Opperman et al. [65] found that stable flies across dairy farms in Germany were 100% resistant to deltamethrin.

### 2.2. Systemic Insecticides

The catalyst to search for a systemic insecticide toxic to insects, but non-toxic to the host was provided when wheat grown in soil containing sodium selenate was not attacked by the aphid *Macrosiphum granarium* Kirby [66]. McGregor and Bushland [67] attributed the first success at internal administration of an insecticide in controlling a bloodsucking insect to Lindquist et al. [68] where rabbits tolerated both DDT and a pyrethrum extract given orally, which resulted in their blood being toxic to bed bugs. Over 10 years, Drummond [69,70,71,72,73,74,75,76,77] reviewed and screened multiple animal systemic insecticides against stable flies, which was first reported in the mid-1950s [67]. Complete mortality of adult stable flies occurred with repeated feedings of 5 ppm of benzimidazole levels in blood [78]. Although many anthelmintic drugs administered to animals leave residues of the drug in the treated animal’s dung where they can impair fly development in the dung, these residues also have both lethal and sublethal effects on non-target dung fauna [79,80].

### 2.3. Insect Growth Regulators (IGRs)

IGRs act by preventing fly larvae from successfully moulting through to the next larval instar by interrupting the synthesis of chitin. Additionally, the juvenilizing activity of ecdysones prevent the adult fly from emerging from the pupal case [81]. Active ingredients that have been used against stable flies include cyromazine [82], pyriproxyfen [83,84], diflubenzuron [85,86], buprofezin [84] and novaluron [87,88]. Addition of these chemicals to either animal bedding, old animal feeding sites, or any larval developmental substrate significantly reduced adult eclosion [89]. The earliest tests using insect growth regulators against stable flies (with juvenile hormone activity) were performed in the 1970s [90,91,92,93,94,95,96,97].

Injection of IGRs into animals such that their urine excretes residues of the drug onto their bedding was also an effective method of control [98]. Application of diflubenzuron to adult resting surfaces in intensive animal production [85] imposed ovicidal impacts on female stable flies, whereby egg hatch was reduced and subsequent development to adult flies inhibited [99]. Bedding from dairy cattle given a bolus of diflubenzuron inhibited stable fly development [100]. Cyromazine, applied as granules to old hay feeding sites, restricted adult stable fly development by ≤97% for up to 10 weeks post-application [101]. Similarly, cyromazine applied as granules to cattle, swine and poultry manure prevented stable fly development for up to 4 weeks post-application [82]. Novaluron applied as granules to cattle feeding sites that contained waste forage, manure, and urine suppressed adult stable fly development by 80%–90% for up to 12 weeks post-application [88]. Recent testing of stable flies across dairy farms in Germany did not show any signs of resistance to the insect growth regulators cyromazine and pyriproxyfen [65]. 

### 2.4. Automatic Sprayers, Backrubbers and Ear Tags

Treatment of animals directly with pesticides involves the use of automatic sprayers [102,103], backrubbers [104,105,106] or insecticide-impregnated ear tags. Excellent control of biting flies was noted with the use of an automatic cattle sprayer, where a small amount of concentrated insecticide (0.2–1.7% pyrethrin) was applied to cattle when they stepped on a hinged platform while going to or from the milking barn or to feed and drink while in pastures [103]. Backrubbers as a method of insecticide application onto cattle was first released by the South Dakota State College Agricultural Experiment Station in the annual reports of 1950 and 1951 [107]. Cable-type back rubbers, soaked with a 5% oil solution of DDT, have been used to control horn flies on rangeland cattle [107]. Due to the effort and cost involved in rounding up cattle, backrubbers were a simpler and cheaper treatment option. Insecticide-soaked burlap sacks were wrapped around a cable slung between two posts that sagged below the height of the animal’s backline in areas where the animals often congregate. Without coercion, the animals walking underneath get a dose of insecticide along their back [106]. Although originally proposed for horn fly control, their use has assisted in efforts to manage stable flies [107].

Several insecticide compounds have been impregnated into plastic ear tags for use in cattle herds as a means of repelling and/or reducing stable flies that attempt to blood feed from the animals. Ear tags containing organophosphate (fenthion) and pyrethroid (flucythrinate, fenvalerate) insecticides have been used in the past with some success (10 weeks control on cattle [58]), limited success [108] and/or no success in reducing the stable fly burden on cattle [109].

### 2.5. Repellents

The use of repellents has become one of the most efficient ways to prevent discomfort associated with insect bites [110]. In 1985, an olfactometer was developed to improve measurement of repellency of chemicals to stable flies [111]. DEET (N,N-diethyl-meta-toluamide) developed in 1944, is considered by many as the gold standard of insect repellents [112]. First used by the military during World War II, DEET became the most extensively used personal arthropod repellent for over 50 years, with a particular focus on repelling mosquitoes. Hundreds of natural products including plant essential oils have been reported demonstrating their insecticidal and repellent properties [113,114,115]. However, nearly all plant-based repellents derived from plant essential oils have limited residual activity (<4 h) [116], primarily due to their high volatility. For example, citronella oil was the first successful plant-based insect repellent, but its effectiveness was limited to only several hours. DEET (>25%) provides up to 10 h of protection against mosquitoes [117].

#### 2.5.1. Repellents on Livestock

Reports on repellents to protect livestock from stable flies first appeared in the 1910s [118,119]. The earliest studies on repelling stable flies from cattle suggested the use of crank case oil and oil of tar as being the most promising results in terms of cost effectiveness and practicality [120]. Repellents specific to stable flies were reported in several studies in the 1940s and 1950s [121,122,123,124]. Most repellent formulations, however, have only demonstrated at best 1 to 2 d reprieve for animals [32,125], and with many less than 12 h of measurable reduction in stable flies [126].

Several essential oils (plant-derived) have been used as repellents against stable flies (e.g., catnip, peppermint, eucalyptus and lemongrass) [127,128,129,130,131,132] but again the repellency only lasts 1–2 d. A repellent oil combination of sunflower (95%), geranium (2.5%) and lemongrass (2.5%) applied to dairy cattle reduced both stable flies on the animals and fly defensive behaviors [133]. DEET has been added to numerous products for fly control to improve the repellency of biting flies. A combination of fipronil and permethrin worked well as a toxicant and repellent towards stable flies on dogs [134]. Research into fatty acids derived from coconut oil has shown excellent repellency to stable flies [135,136,137] with up to 2 weeks repellency being demonstrated in laboratory bioassays, and up to 96 h protection when applied to cattle in the field [138].

#### 2.5.2. Repellents on Humans

The use of topical repellents to stable flies for humans was assessed by Gilbert et al. [139] where 4-pentyl-2-oxetanone was more repellent than DEET. Diethylphenyl-acetamide was identified as a new and safe repellent of stable flies when tested on rabbits [140], with its use then proposed for humans. Personal protective equipment to reduce the impact of biting flies (including stable flies) for both military [141] and civilian personnel typically involve rinsing or impregnating clothing with permethrin [142,143,144]. 

## 3. Biological Control

### 3.1. Natural Predators of Stable Flies

Since the last reviews of natural enemies of stable flies [145,146], there have been numerous papers on predators of stable flies, either from naturally occurring larvae and pupae and/or sentinel pupae set up in intensive livestock production areas. Smith et al. [147] noted that field mortality of stable flies was due to predation in their larval development sites. Hall et al. [148] found 19 arthropod species in five families to be predaceous as immatures or adults, in association with stable fly immatures developing in grass clippings in central Missouri, US. This study indicated that between 34% and 73% of apparent mortality was due to predation of stable fly eggs and larvae. Wasp parasitism in the field was as high as 13%–20% for stable fly and house fly pupae in Californian dairies [149], 10.6% of stable fly pupae from cattle feedlots [150] and 10% of stable fly pupae from Danish dairy farms [151]. All natural predators of stable flies recorded in the literature are listed in Table 1 (hymenopteran parasites) and Table 2 (non-hymenopteran insect predators).

By contrast, Floate et al. [161] found that <1% of stable fly pupae in the field were parasitised. McKay and Galloway [165] found that 3.8% of naturally occurring house fly and stable fly pupae were parasitised by nine different wasp species. Further, the authors concluded that releasing >3.5 million *Nasonia vitripennis* wasps did not result in substantial parasitism in either sentinel pupae or naturally occurring pupae of both fly species. Many other studies have found natural parasitism rates below 5% [150,160,162] and most often less than 1% [149,151,152,153,165]. Total filth fly parasitism of natural pupae collected from cattle feedlots in Alberta, Canada did not exceed 7% [168].

#### 3.1.1. Wasps

The Pteromalidae family of wasps are only 2–3 mm in size [169] and harmless to humans and livestock. However, many are insect parasitoids and several species are important biological control agents of pest insects, in particular filth fly species associated with intensive animal production. The adult wasp lays one or more eggs on the surface of the fly pupa inside and the developing wasps kill the host fly. Fly parasitoid wasps associated with livestock production in North America have been widely studied, including species in the genera *Muscidifurax*, *Spalangia*, *Trichomalopsis*, *Nasonia*, *Urolepis* and *Pachycrepoideus*. Within and between each genera, there are differing habitat choice, host choice, and behavior. Some wasp species live gregariously as larvae within a host puparium, while most species are solitary as immatures. Similarly, some species prefer constantly wet habitats such as manure, whereas others prefer drier areas such as manure-soiled straw bedding. The abundance and habitat preference of each parasitoid species can differ between regions and seasons. 

#### 3.1.2. Staphylinids

Staphylinidae beetles appear almost wingless and are known to predate on fly eggs and larvae in association with either cadavers or rotting plant material [176,177,178]. Smith et al. [146] noted that field mortality of stable flies was principally due to predation by staphylinid beetles. Staphylinids have been claimed to suppress biting fly populations (including mosquitoes) by Frank and Thomas [179], but without any supporting evidence. Staphylinids recorded in the literature as predators of stable flies are listed in Table 2. *Aleochara* species were found parasitising filth fly pupae in Kansas [171], but <0.8% of 22,000 stable fly pupae collected from cattle feedlots produced any adult beetles (mostly *Aleochara lacertina* and some *Aleochara bimaculata*). Similarly, *Aleochara puberula* only emerged from 1% of stable fly pupae in Brazil [172]. 

The author has seen staphylinid beetles (*Aleochara* spp.) feeding on stable fly eggs in rotting celery residues left after harvest near Perth, Western Australia. Adults and larvae of some *Philonthus* species occur in dung of ungulates and eat fly eggs and larvae [179]. When grass clippings were used as an artificial breeding medium for stable flies, the staphylinds *Philonthus americanus* and *Oxytelus sculptus* were the most abundant insect predators [158]. 

#### 3.1.3. Mites

Mites were first touted as a biocontrol agent of synanthropic flies by Axtell [180]. Only two studies have looked at predation on stable fly eggs by mites. *Macrocheles muscaedomesticae* (Acarina: Macrochelidae) had negligible impact on stable flies in Kinn [174], despite it being noted that this mite was the most consistent and abundant predator to arrive early at sentinel, wet, ryegrass clippings seeded with laboratory-reared stable fly eggs [146]. *Macrocheles embersoni* was the best predator out of three mite species assessed against stable fly eggs and larvae, consuming 24 larvae per day [181]. Stable flies carrying mites represent only 5%–10% of the population at dairy and beef farms, with the mites major effect being to limit their dispersal compared with mite-free stable flies [175]. 

#### 3.1.4. Nematodes

In addition to the arthropod predators identified in Table 1, nematodes from the Mermithidae family were found predating on stable flies [145]. Entomopathogenic nematodes search for and locate their hosts by detecting products of excretion, CO_2_ levels and temperature gradients. When a host is located, they penetrate through natural openings (mouth, anus and spiracles) or actively penetrate through the cuticle [182]. They then migrate into the hemocoel of the host [183] where their symbiotic bacteria release toxins that kill the host [184]. Nematodes were first noted as infecting stable flies by Poinar and Boxler [185] and further examined under laboratory conditions [186]. The use of entomopathogenic nematodes to control stable flies developing at round bale feeding sites has been tested by Pierce [187], who found that nematodes in the genera *Steinernema* and *Heterorhabditis* all showed efficacy against stable fly larvae. The commercially available strain *Steinernema feltiae* produced the highest mortality in stable flies (56%) in laboratory bioassays on a hay/manure substrate [187]. Leal et al. [182] demonstrated that *Heterorhabditis* spp. of nematodes show potential against stable fly larvae, but they have not been used in any commercial context to control stable flies to date. 

#### 3.1.5. Microsporidia

Microsporidia are spore-forming unicellular parasites that were considered protozoans or protists, but are now known to be fungi, or a sister group to fungi. Stable flies were not susceptible to the microsporidian parasite *Octosporea muscae domesticae* [188].

### 3.2. Commercially Available Predators used to Control Stable Flies

A number of commercially available parasitic wasps have been used against stable flies, mostly in intensive animal industry settings and/or animal stables. Augmentative biological control of filth flies has become a commonly used technique for fly control on equine facilities [189], cattle feedlots [190] and poultry production [191]. Given suitable habitats and hosts, commercially available pupal parasitoids have the potential to suppress populations of filth flies. Studies on livestock facilities with various species of wasp have yielded mixed results, with suppression of fly populations from augmentative releases of parasitoids in some situations [192,193,194,195,196,197,198] and failures in others [165,199,200,201,202,203].

Releases of *S. endius* on cattle feedlots over 13 weeks did not significantly increase parasitism of stable fly pupae [204]. The fly pupal parasitoids *M. raptor*, *M. zaraptor* and *S. nigroaenea* from commercial insectaries failed to significantly reduce stable fly numbers despite weekly releases of high numbers at a cattle feedlot and dairy [199,202]. Weekly releases of *S. nigroaenea* over 4 months for 3 years on cattle feedlots in Illinois, US increased stable fly mortality [203]. In pig premises in Norway, bi-weekly releases of *S. cameroni* suppressed stable flies [205]. Weekly releases of *S. cameroni* on dairy and pig farms in Denmark did not affect stable fly numbers [198], whereas bi-weekly releases of the same wasp on dairy farms in Denmark did suppress stable fly numbers on cattle [206] and on one of two pig farms in a study conducted in Norway [205].

These contradictory results may be partially attributed to environmental factors that affected parasitoid abundance and distribution [198]. Study specific factors such as the use of insecticides, the use of low-quality commercial colonies, parasitoid microhabitat preferences, the availability of hosts, methods of release and timing of release may have also affected their effectiveness [207,208]. In addition, it is possible that existing parasitoid populations may inhibit the establishment or function of newly released species in augmentation programs [209]. Quarles [210] suggested that the success of a biological control program using pupal parasitoids relies on matching the released species with the climate and habitat of the release location, most appropriately by deploying endemic species.

### 3.3. Biopesticides

#### 3.3.1. Plant-derived Products

Many plant-derived products with insecticidal activity against stable flies have been found. The first example in the literature was when >70% of stable fly larvae died when reared on media containing 800 ppm of L-canavanine. This compound is a non-proteinogenic amino acid found in some legumes that acts as a toxin in the plant to protect against insect attack [211]. A soybean trypsin inhibitor was encapsulated in bovine red blood cells and fed to adult stable flies, resulting in 50% mortality and elimination of egg production [212]. The plant flavonoid pinocembrine exhibits larvicidal activity against stable flies [213], whilst extracts from Chaste tree seeds (*Vitex agnus castus*) effectively repelled biting flies for ≈6 h [214]. An extract from *Melinis minutiflora* grass also had pesticidal activity against stable fly adults and larvae [215].

#### 3.3.2. Semiochemicals

Semiochemicals are signalling compounds that carry information between individuals of a species and cause a change in their behavior such as attraction or repellency [216]. Semiochemicals have proved to be highly potent insecticides against stable flies. Compounds such as rosalva, citronellol, geranyl acetone [217], beta-damascone, cyclemone A and melafleur [218] show remarkable toxicity to stable flies in both the laboratory and under field conditions.

#### 3.3.3. Entomopathogenic Fungi

Entomopathogenic fungi are part of the decomposer community and inhabit soils where many filth fly species spend most of their life as mobile larvae or dormant pupae. Many fungi have developed the ability to invade different life history stages of flies, and in doing so, use the insect host to complete their life cycle. In a 2007 review by Wraight et al. [219], there were >700 species of fungi that were lethal to insects (i.e., entomopathogenic). Most of the important entomopathogenic fungi produce asexual non-motile spores or conidia, which are capable of surviving for long periods (1–5 yrs) during unfavourable environmental conditions (typically drought). Under favourable conditions, the conidia become infective units, which contact, germinate and pentrate the insect cuticle where they invade the hosts body to later release spores when the insect dies. The use of entomopathogenic fungi against stable flies represents an opportunity to deal with a serious pest of livestock and humans by biological means as compared with chemical intervention.

Examples of specific entomopathogenic fungi impacting stable flies include *Metarhizium anisopliae, Metarhizium brunneum, Lecanicillium lecanii (formerly Verticillium lecanii), Beauvaria bassiana, Entomophthora muscae* and *Entomophthora schizophorae*. Specifically, Moreas et al. [220] showed that a strain of *M. anisopliae* killed all stable fly eggs in a laboratory bioassay, but did not have any impact on larvae or pupae. This can be explained by the proven antimicrobial activity of stable fly larvae against other entomopathogenic fungi [221,222]. López-Sánchez et al. [223] showed that 7 days after exposure to isolates of *M. anisopliae* and *B. bassiana* collected from the soil of lodging pens of dairy production units, mortality of adult stable flies was >90%. *Metarhizium brunneum* was shown to significantly reduce stable fly oviposition when substrates were treated with two commercial fungal products [154]. González [224] suggested that accelerated decomposition of pineapple crop residues in Brazil could cause the material to be less attractive to female stable flies. Paganella-Chang [225], however, found that the rate of decomposition of pineapple stubble was not responsible for the control of stable flies, but rather the composition of the fungal decomposer community (i.e., *Acremonium, Fusarium* and *Trichoderma* spp.). Weeks et al. [226] demonstrated that commercial fungal agents containing *M. anisopliae* and *B. bassiana* induce mortality in adult stable flies.

Strains of *B. bassiana* have been used in several studies against larval and adult stable flies. Up to 90% mortality was demonstrated in adult stable flies exposed to *B. bassiana* [223]. Moreas et al. [227] showed pathogenicity of *B. bassiana* to immature stages of stable flies. Watson et al. [228] showed that *B. bassiana* helped control stable flies in the sawdust bedding in calf hutches, whilst Oliveira et al. [229] demonstrated the use of this fungus in controlling insect pests in poultry production. *Verticillium lecanii* and *V. fusisporum* were found to be pathogenic against stable flies where the fungus was found in adult populations in Denmark [230]. There are some fungi that clearly have species-specific effects on different nuisance flies. For example, 100% of house flies and only 2% of stable flies were susceptible to infection by *E. muscae* [231]. The fungus *L. lecanii* had no effect on immature stages of stable flies [232]. In Mexico, several applications of *M. anisopliae* formulations to cattle (4 times over 21 days) reduced stable fly numbers by 73% and reduced defensive fly behaviors by ≈67% [233].

#### 3.3.4. Entomopathogenic Bacteria

Mortality was only induced in adult stable flies that fed on an isolate of *Bacillus thuringiensis* (*thompsoni* 4O1) and only when fed via blood or applied topically [234]. Similarly, Gingrich [235] showed that stable flies were the most resistant dipteran pest of cattle to *B. thuringiensis* when this bacterial agent was added to the feed of cattle.

## 4. Sterile Insect Technique (SIT)

The use of the Sterile Insect Technique against stable flies was only briefly pursued in the 1970s and early 1980s [236,237]. A feasibility study by LaBrecque et al. [238] into the release of sterile male stable flies to control wild population of stable flies on a small island (218 km^2^) in the Caribbean Sea (St. Croix, US Virgin Islands) showed that control was possible during the dry season using the mass rearing and sterilization outlined in Williams et al. [239]. Patterson et al. [240] reported that after 1 y of releasing 0.5 million sterile males each week on St. Croix, the wild population of stable flies across the island was reduced by 99.9%.

The traditional SIT method utilizing irradiation has not been pursued further for controlling stable flies due to three key aspects of this pest’s biology outlined in Taylor [4] and summarised here. Firstly, both sexes blood-feed several times per day, so any release of flies would significantly increase the impact on livestock nearby; secondly, stable flies disperse over a great distance, meaning that very high numbers would need to be released to affect control. Finally, stable fly outbreaks can rapidly develop due to their ability to exploit numerous substrates for larval development and their high reproductive rate.

## 5. Physical Control

### 5.1. Trapping

Stable fly trapping systems to catch and remove stable flies from areas where they are seriously affecting livestock (e.g., barns, stables, stalls, water troughs) form a large part of physical control. The first recorded trap for catching stable flies was by Hodge [241], where a large trap was fitted into a stable window and all other windows into the stable were covered. The trap had no bait and relied on the attraction of adult stable flies to the light coming through the trap apparatus. Over 4 months, this trap caught ≈ 4.3 L of stable flies (number unknown). The use of fly traps was outlined by Bishopp [242] for attracting and catching houseflies, blowflies and stable flies. Early versions of stable fly traps consisted of either emergence, animal-baited [243] or box-type traps with shingles or panels coated with an adhesive [244,245].

#### 5.1.1. Surveillance/Monitoring Traps

Two key traps for monitoring stable flies have been subsequently modified and expanded upon: (1) the Williams trap using panels of the fiberglass product Alsynite [246] and (2) the use of Alsynite in cylindrical traps [247,248,249,250]. Clear fiberglass and ultraviolet reflectors are the most efficient material for trapping stable flies, especially when exposed in full sunlight [251]. The attractiveness of the spectral properties of Alsynite to stable flies was verified by Agee and Patterson [251]. More recently, plastic sheeting in the form of Coroplast^®^ (Great Pacific Enterprises., Granby, Quebec, Canada) and polyethylene terphthalate have been shown to be more attractive to stable flies than Alsynite [20,252].

#### 5.1.2. Electrocutor Traps

The first use of an electrified grid was reported by Wells [253], where its use against stable flies was only loosely reported in a minor experiment with no fly counts given. Electrocutor grids re-appeared in the 1970s, with Morgan et al. [243] showing that they caught three times as many female stable flies as cage traps. Shreck et al. [254] showed that in combination with carbon dioxide, electrocutor grids were a highly selective, stable fly killer. A solar-powered electrocuting grid killed 4000 and 1200 stable flies/day in two studies [255,256]. One negative aspect of electrocuting insect traps is that they can release bacterial pathogens into the air such as *Serratia marcescens* and *Escherichia coli* and potentially spread infectious disease agents [257].

#### 5.1.3. Walk-Through Traps for Livestock

In 1930 Loughnan [258] first investigated the concept of a walk-through fly trap for cattle with biting flies (horn and stable flies) being dislodged by a brush of leaves and branches as they walked into a darkened and partitioned building. Although principally developed to remove horn flies, this trap was suggested by Segal [259] as being more effective against stable flies. Later Hall and Doisy [260] tested a more portable walk-through trap developed by Bruce [261] for horn fly control, which Hall and Doisy [260] considered was better suited to stable fly control. A vacuum device was added to a walk thought trap by Denning et al. [262], which sucked up horn flies off dairy cattle as well as stable flies. This trap however only removed on average 10 stable flies/animal/walk through event [262].

#### 5.1.4. Modification of other Biting Fly Traps

Traps designed to catch other biting flies that bother livestock including tabanids and horn flies also caught stable flies whose numbers were often recorded [263,264,265,266,267,268,269]. These include the Nzi [266], Vavoua [270], and Manitoba horse fly trap [271]. Several standard traps have been modified in terms of their colour [272], contrast with the trap border [273], size, non-drying adhesive on the trap surface [272], and odour source [274]. Catch rates of stable flies improved using dry-ice baited traps that were developed for capturing mosquitoes [275,276,277]. Release of CO_2_ at 3 L/min caught three times as many stable flies compared with no gas or carbon monoxide [278]. Dry ice and components of animals breath such as 1-octen-3-ol [279] and acetone [280] increased catches of *Stomoxys* spp. [264].

Pickens [281] provided a review of the different trap types and attractants developed to catch stable flies, including trap colour, orientation and design. The effect of trap colour and type of adherent on catches of stable flies were assessed by Ruff [282] with blue traps catching more flies. Variations on the Williams trap were tested by Scholl et al. [283]. Different coloured beach balls and plasticized corrugated boards were assessed on catching stable flies along Florida beaches [284,285]. The ideal configuration, size and colour of target traps for stable flies was assessed by Hogsette and Foil [286]. Of late, the Knight Stick Trap and Sticky Wraps have been shown to be useful in areas where animals are housed and/or where pesticides are not allowed [287,288].

#### 5.1.5. Comparative Trap Studies

Broce [249] improved on the Alsynite trap by using a cylindrical design (30 cm diameter x 30 cm high), which caught more stable flies/unit area than the Williams trap [249,250]. Most stable flies notably land on the side protected from the wind [249,289]. Taylor and Berkebile [20] compared six different stable fly traps where a prototype trap with multi-sided, clear plastic panels and coated in a non-drying glue was the most efficient at trapping stable flies. Alsynite cylinder traps caught four times the number of stable flies as blue-black cloth targets around cylindrical traps [290]. Both trap types reflect and refract ultraviolet light, which attracts stable flies. Studies comparing various traps in catching stable flies were conducted by Mihok et al. [264] and Gilles et al. [291].

#### 5.1.6. Toxic Traps

The concept of an attractant trap surface, material (e.g., Alsynite) or fabric being used as a toxicant or control system for stable flies was first proposed by Meifert et al. [292] who suggested treating Williams trap panels with a synthetic pyrethroid. Using this system, the authors reported a reduction in the local population of stable flies by ≈ 85% after 1 weeks trapping (the trap units removed >30% of the adult population/day). Tseng et al. [293] suggested wrapping the panels on a Williams trap with white yarn impregnated with permethrin, as most pesticides easily wash off the impervious fiberglass. When the yarn is wrapped in a continuous coil (1.3 cm apart), it maximised contact by stable flies. Field longevity of the permethrin-impregnated yarn was estimated at 6–8 weeks [294]. Extrapolation of this concept to cloth targets treated with pesticide was first suggested by Foil and Younger [295]. They demonstrated that six times more stable flies were attracted to cloth targets compared with Alsynite traps. Fabric targets treated with 0.1% λ-cyhalothrin remained effective against a susceptible strain of stable flies for up to 3 months exposure in the field in the US [296]. A recent laboratory bioassay study in Brazil showed that when stable flies had 30s contact with fabric cloths impregnated with 1% fipronil all the flies were dead when assessed 24 h after exposure; stable flies had to contact fabric impregnated with 1% chlorpyrifos for 60s for >98% of flies to be dead 24 h later [297].

#### 5.1.7. Effective Stable Fly Control through Trapping

Sticky traps, strips and ribbons have been used to monitor stable flies in different environments [298], but in terms of reducing their population numbers, there are few conclusive studies. One notable exception was by Rugg [299], where the Williams trap [246] was shown to reduce numbers of stable flies at a zoo in Australia by 79% after 1 week (26% of population removed/day). Similarly, Alsynite cylinder traps caught 80% of stable flies within a zoological park in the US compared with a blue-black cloth target trap modified into a cylindrical trap [290]. Pickens [281] reviewed the use of traps to suppress stable fly populations. Sticky pyramid traps [300] and pyramid traps treated with tralomethrin (SP) [301] reduced stable fly populations on dairy farms. Large sticky traps in dairy calf facilities reduced stable flies by ≈14,000 flies/week, which producers felt kept the numbers at acceptable levels [302]. Tam et al. [303] found that sticky traps did not protect horses from stable fly blood feeding. Several studies have used trap counts of stable flies to model their population dynamics [304,305,306] where the effects of weather on capture of stable flies has been quantified [248,307]. 

### 5.2. Physical Protection of Livestock

Horse owners often put protective rugs, fly boots and face masks on their horses to reduce the numbers of stable flies being able to blood feed from their animals. The use of mesh leggings and leg bands as a physical barrier (non-insecticidal) to stable flies reduced foot stomps behavior in both horses [308] and dairy cows (only leggings tested) [309]. A recent and novel approach has seen the painting of black and white stripes on livestock such as cattle, which can reduce biting fly attacks (mostly stable flies) and associated fly-avoidance behaviors [310]. 

### 5.3. Physical Barriers

In vegetable production areas north of Perth, Western Australia, post-harvest residues can support the development of >1000 stable fly adults/m^2^ [2]. A novel and pesticide-free approach to controlling stable flies in this setting was developed by Cook et al. [311], which involved the burial of post-harvest residues followed by the compaction of the sandy soil above the residues. This resulted in a hard crust forming in the sand, that newly-emerged stable flies could not dig their way through to emerge at the soil surface [311]. This non-chemical method of control provided benefits to vegetable producers, including the retention of organic matter from the residues, less soil wind erosion and no need for a pesticide application. The use of plastic covers on pineapple crop residues increased the rate of anaerobic decomposition of the residues and prevented stable flies from accessing the residues. With increasing time under plastic cover, the pineapple stubble attracted significantly less stable fly oviposition, decreasing from 2928 eggs/m^2^ after 10 days cover to just 152 eggs/m^2^ after 30 days cover [312]. Within animal housing and dairies, air curtains produce a physical barrier to flies and can help to reduce their numbers on animals [313].

## 6. Cultural Control

Cultural controls are typically the oldest methods that have been used to manage insect pests. However, with the development of cheap synthetic pesticides, these controls were either abandoned or considered too costly compared to using pesticides [314]. Because cultural controls are preventative rather than curative, they are dependent on long-range planning and require a thorough understanding of the biology and ecology of the target pest. Cultural control against stable flies in this review examines all the hygiene and sanitation measures used to either remove or alter the substrates that stable fly larvae can develop within.

Greene [315] stated that sanitation by the removal of animal and plant residue to prevent fly development is the most important and first method of fly control to be used for fly reduction. This simple method of stable fly control remains highly relevant for intensive animal production and accumulation of rotting plant residues in horticultural production. Reducing stable flies and their effects on livestock without the use of pesticides was reviewed by Pickens et al. [316]. The major benefits of cultural control methods are less selection pressure for insecticide resistance and less chemical residues in the environment that may negatively impact beneficial, non-target organisms.

### 6.1. Sanitation

Regular sanitation schedules, where stable fly oviposition substrates and larval developmental sites are removed, have been shown to be effective in reducing stable fly adult populations [317,318,319,320]. Bishopp [321] noted that stable fly numbers can be kept down by proper handling of stable refuse, either by stacking or otherwise disposing properly of any accumulations of straw or hay, especially adjacent to stables. These sanitation efforts remain highly relevant wherever stable flies continue to affect livestock and are often the simplest and cheapest method of stable fly control. Uncovered silage stacks encourage stable fly development [322], which when simply covered was shown to be an effective means of preventing stable flies breeding [323].

### 6.2. Manure Amendments

Moore et al. [324] reviewed chemical amendments that would prevent ammonia volatilization from poultry litter. Addition of organic acids to the bedding material used in meat chicken production reduces the litter’s pH and prevents the release of ammonia, which is a general fly attractant and oviposition stimulant [325]. For example, incorporation of sodium bisulfate reduces ammonia levels and stable fly populations in horse barns and calf hutches [326,327]. Calcium cyanamide (1%–2.5% v/v) and sodium bisulphate (10%) reduced the numbers of stable flies that developed from raw poultry litter by as much as 99%–100% [328]. In addition, the high nitrogen fertiliser calcium cyanamide was highly toxic to stable fly larvae both in the laboratory [329] and in the field when applied to a medium consisting of feedlot and dairy barn waste [330].

### 6.3. Animal Bedding

Several bedding treatments and options have shown a reduction in the number of stable fly larvae developing in animal bedding as their manure and urine mixes in with the bedding material. For example, sawdust, wood chips and ground corncob bedding significantly reduced densities of stable fly larvae in calf hutches [98,331]. The authors suggest that the aforementioned substrates do not support stable fly larval development as they do not have a high moisture content, are not rich in organic matter and do not promote microbial activity [98,331], which is essential for stable fly larval survival and development [332].

## 7. Integrated Pest Management (IPM)

IPM of stable flies in general promotes the use of surveillance and monitoring of stable fly numbers to guide the use of chemical, biological and cultural control options in such a way that pesticide use is minimised. This will (i) allow for the use of commercially available biological control agents, (ii) encourage beneficial insects and natural predators of stable flies, and (iii) reduce the selection pressure for resistance to insecticides. Sanitation or on-farm hygiene are the simplest methods to employ, which includes regular removal of animal manures and soiled animal bedding. In addition any piles of plant material that can rot and support the development of stable fly larvae need to be either removed, buried under at least 1m of soil, or buried and the soil above the plant material compacted [311]. IPM plans rely on surveillance and monitoring of fly pests to enable the correct identification of the target pest species and Kaufman et al. [333] provides a list of common pest flies including stable flies in urban and rural environments. IPM in urban environments was promoted by Greene [315] using a range of chemical, cultural and mechanical methods to control both stable flies and house flies.

IPM programs are particularly useful in intensive animal industries (dairy, beef, swine, and poultry) [334] for nuisance fly control and some specifically for managing stable flies. Integrated control strategies have been promoted for stable fly control associated with poultry [335,336], beef cattle [337,338] and dairy cattle [300,338,339,340,341,342]. IPM of stable flies requires an area-wide approach to its management as the adult flies can travel long distances and their larval developmental sites are “diverse, dispersed and often difficult to locate” in agroecosystems [4]. For example, a program to reduce stable flies developing in the agricultural areas of western Florida was initiated to reduce the large, migratory swarms of stable flies to beaches and holiday resorts in Florida, which damaged the local tourism industry [343].

## 8. Conclusions

This review represents a comprehensive overview of all chemical, biological and cultural control management strategies employed over more than a century against stable flies, a continuing global pest of livestock and humans. This paper should serve as a complete and exhaustive reference for entomologists, livestock producers and horticulturalists with any involvement in management of stable fly populations, either from a research or technical perspective.

## Figures and Tables

**Table 1 insects-11-00313-t001:** Natural hymenopteran parasites of *Stomoxys calcitrans* recorded in the literature.

Hymenoptera (Wasps)	Family	Reference
*Muscidifurax raptor* Girault and Sanders	Pteromalidae	[146,152]
*Muscidifurax zaraptor* Kogan and Legner	Pteromalidae	[146,152]
*Pachycrepoideus vindemiae* Rondani	Pteromalidae	[152]
*Spalangia cameroni* Perkins	Pteromalidae	[149,153,154]
*Spalangia drosophilae* Ashmead	Pteromalidae	[155]
*Spalangia endius* Walker	Pteromalidae	[149,150]
*Spalangia haematobiae* Ashmead	Pteromalidae	[152,156]
*Spalangia nigra* Latrielle	Pteromalidae	[157]
*Spalangia nigroaena* Curtis	Pteromalidae	[150,158]
*Spalangia subpunctata* Förster	Pteromalidae	[152]
*Trichomalopsis dubius* Ashmead	Pteromalidae	[159]
*Trichomalopsis viridescens* Walsh	Pteromalidae	[152]
*Urolepis rufipes* Ashmead	Pteromalidae	[160,161,162]
*Nasonia vitripennis* Walker	Pteromalidae	[163]
*Dibrachys cavus* Walker	Pteromalidae	[159,161]
*Aphaereta pallipes* Say	Braconidae	[152,162]
*Rubrica surinamensis* DeGeer	Bembicidae	[164]
*Diplazon laetatorius* Fabricius	Ichneumonidae	[156]
*Phygadeum fumator* Gravenhörst	Ichneumonidae	[151,165]
*Trichopria stomoxydis* Huggert	Diapriidae	[166]
*Trichopria* spp.	Diapriidae	[167]
*Coptera* spp.	Diapriidae	[167]

**Table 2 insects-11-00313-t002:** Natural non-hymenopteran insect predators of *S. calcitrans* recorded in the literature.

	Family	Reference
**Coleoptera (Beetles)**		
*Aleochara bilineata* Gyllenhal	Staphylinidae	[170]
*Aleochara bimaculata* Gravenhörst	Staphylinidae	[171]
*Aleochara lacertina* Sharp	Staphylinidae	[158,171]
*Aleochara puberula* Klug	Staphylinidae	[172]
*Belonuchus rufipennis* Fabricius	Staphylinidae	[145]
*Lithocaris ardenus* Sanderson	Staphylinidae	[145]
*Oxytelus sculptus* Gravenhörst	Staphylinidae	[158]
*Philonthus americanus* Erichson	Staphylinidae	[158]
*Philonthus brunneus* Gravenhörst	Staphylinidae	[145]
*Philonthus hepaticus* Erichson	Staphylinidae	[145]
*Philonthus rectanguulus* Sharp	Staphylinidae	[145]
*Philonthus sericans* Gravenhörst	Staphylinidae	[145]
*Philonthus theveneti* Horn	Staphylinidae	[173]
*Staphylinus maculosus* Gravenhörst	Staphylinidae	[145]
**Acarina (Mites)**		
*Macrocheles muscaedomesticae* Scopoli	Machrochelidae	[146,174]
*Macrocheles subbadius* Berlese	Machrochelidae	[175]

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
