# Peer review of "A Historical Review of Management Options Used against the Stable Fly (Diptera: Muscidae)"

_insects, 2020, doi:10.3390/insects11050313_

Round 1
Reviewer 1 Report
The revised review by Cook flows well and presents the information in easily n and navigable and relevant sections. Inclusion of the section on trapping, while not required, wwas welcome.
Some minor suggested edits below, but otherwise acceptable for publication.
Citations within main text:
L120: Similarly, Reissert-Opperman et al [63]...
L363: Fix ciations to remove Author in list
L363: A feasability study by Author [232]...
L395: Author [238]
L480: [299-301]
L481: [245, 302]
L136, L137: Remove apostrophe (IGRs)
L179-180: A soybean trypsin inhibitor was encapsulated in bovine red blood cells and fed to adult stable flies, resulting in 50% mortality and elimination of egg production.
L202: repellency towards?
L203: worked well as a toxicant and repellent towards stable flies on dogs
L227: delete second 'parasitized.'
L258-259: Reference for Philonthus sentence?
L266: delete period after Machrocheles
L271: Do you mean 'byproducts'?
L284: Stable flies were not susceptible
L328: Examples of specific entomopathogenic fungi impacting stable flies includes
L328-343: italicize scientific names
L337-340: 'less attractive to female stable flies, but Paganella-Chang...control stable flies. Rather, the composition...'
L367: Define 0.5m?
L369: The traditional SIT method, utilizing irradiation, has not been pursued further for controlling stable flies due to...
L376: manage pests; however,
L390: methods are
L390-391: less selection pressure...and less chemical residues in the environment that may impact beneficial, non-target organisms.
L419: Loughnan [255][ first investigated the concept...
L449: 4 times
L452: Mihok et al (there are 3 authors)
L463: 6 times
L467: 24 h
L485: '...stable fly numbers can be minimized by proper...'
Author Response
The revised review by Cook flows well and presents the information in easily and navigable and relevant sections. Inclusion of the section on trapping, while not required, was welcome. Some minor suggested edits below, but otherwise acceptable for publication.
Here are my responses to Reviewer 1’s suggested edits and changes
L120: Inserted the authors name Reissert-Opperman et al ...
L363: Removed “LaBrecque et al. 1975" and [24] citation which was incorrect
L363: Inserted the authors name “LaBrecque et al.” before the reference citation
L395: Inserted the authors name “Hodge” before the reference citation
L480: changed citations so it now shows [299-301]
L481: changed citations so it now shows [245, 302]
L136 removed apostrophe from (IGR’s)
L137 Remove apostrophe from IGR’s
L179-180: Added extra detail so the sentence now reads “..A soybean trypsin inhibitor was encapsulated in bovine red blood cells and fed to adult stable flies, resulting in 50% mortality and elimination of egg production…”
L202: The sentence already states that the repellency is towards biting flies
L203: changed the text so it now reads “..worked well as a toxicant and repellent towards stable flies on dogs…”
L227: deleted the second “parasitized”'
L258-259 Added the reference Frank and Thomas (2008)
L266: deleted the period after “Machrocheles”
L271: I did not mean “byproducts”, so changed the sentence to now clarify “by detecting products of …”
L284: Changed the text to now read “Stable flies were not susceptible…”
L328: The first sentence of the paragraph now reads “Examples of specific entomopathogenic fungi impacting stable flies includes….”
L328-343: italicized the scientific names of all the fungal species
L337-340: Changed the text to now read less attractive to female stable flies. Paganella-Chang [219] however, found that the rate of decomposition of pineapple stubble was not responsible for the control stable flies, but rather the composition of the fungal decomposer community …”'
L367: Defined 0.5m as 0.5 million
L369: Changed the text to now read “The traditional SIT method, utilizing irradiation, has not been pursued further for controlling stable flies due to....”
L376:Changed text to now read “..manage pests. However,…”
L390: changed text to read “methods are…”
L390-391: changed the text to now read “less selection pressure...and less chemical residues in the environment that may impact beneficial, non-target organisms…”
L419: Inserted the authors name “Loughnan” before the reference citation
L449: changed to now read “four times”
L452: changed to “Mihok et al “
L463: changed to now read “six times”
L467: inserted a space between “24” and “h”
L485: changed the text to now read “..stable fly numbers can be minimized by proper...”

Reviewer 2 Report
I greatly appreciate the additions and I think they bring a lot to the paper.
I suggest that trapping is moved into a new section, either surveillance/monitoring or physical control, and I suggest that SIT is moved into its own section, perhaps called "new technologies...future advances" you could add any additional items that fall into that category there. I suggest that biopesticides are combined with biological control, and various other suggestions for changing the order, highlighted in the attached PDF.
There is still a lot of places where the authors names need to be added in front of the numbers, I have highlighted them.
I have made lots of additional comments in the attached PDF.

Author Response
Please refer to attached file, as the responses are too long to insert here

Reviewer 3 Report
The paper looks good now, all my earlier concerns have been addressed.
Author Response
Reviewer3 only had the comment
"The paper looks good now, all my earlier concerns have been addressed"
This reviewer had no changes/edits to the manuscript
This manuscript is a resubmission of an earlier submission. The following is a list of the peer review reports and author responses from that submission.
Round 1
Reviewer 1 Report
The review of control options against stable flies provides a welcome historical context for current researchers invested in supplementing existing and developing alternative or new control technologies for this livestock pest.
The current version requires minor revision.
Numerous references are incomplete. A number are missing journal names and pages, while others are in the incorrect format.
Throughout, manuscript would read better if '[#]' format was replaced by 'Author, Year' format at beginning of sentences or when including a callout to author. For example, L60: "...was by Beach et al. (1904; [60])..." rather than "...was by [60]...". Other instances occur in L64, L191, L249, L376, 391. All are not listed here.
L49-50: Awkward. Considering re-wording. "...control options from a historical perspective and supports any entomologist, livestock..."
In the intro paragraph to discussion of insecticides, it would be helpful to frame stable fly biology, as insecticide application is less effective given they don't spend as much time on host as other ectoparasites, like horn flies.
L70: Sentence starting "Between 1935..." is awkward.
L76: Sentence starting "DDD..." is incomplete.
L83: "...[75,76]. Gas condensate, produced by....coal, was tested..."
L93: Sentence starting "DDT and chlordane..." awkward. Need to resolve clauses in parentheses.
L104: "...synthetic pyrethroids (SP),..." so that reference in L108 makes sense.
L108: "...cattle and observed effective control..."?
L114: Also recent 2019 report in Germany.
L132: "Insect growth regulators, such as..."
L141 and 149: delete period before [#]
L161: "...in reducing..."
L175: delete 'very'
L182: DEET
L191: Sentence starting with "In 1985..[162]" would be more relevant in Section 2.6 because it's a tool used for evaluating the repellent phenotype.
L192: Does this reference [163] include testing on humans?
'Semiochemical' is a general term for volatile or non-volatile compounds that modify behavior, including attractants and repellents. As such, the information in section 2.9 can be incorporated into sections above. These compounds can have both repellent and toxicant effects.
L197: "...potent and provide avenues as new insecticides for use against stable flies."
The introductory paragraph for biological control (Section 3.1) only sets up wasp parasitoids, and there is extensive description of pteromalid wasps continued in the subsequent paragraph. Would be helpful to include more in intro about the other predators or reduce length of wasp description.
Section 3.2: Consider editing the section title to place more emphasis on the difference between sections 3.1 and 3.2: 3.1 is NATURAL predators and 3.2 is COMMERCIALLY AVAILABLE predators.
Section 3.2 needs to start with a segue paragraph on these commercially available approches. For that reason, the section would benefit from re-ordering paragraphs 1 and 2 (Paragraph 1 would be L285-L290, paragraph 2 would be L272-284).
Section 4.3: If there are going to be separate sections on repellent versus insecticidal activity of plant derivatives, then references to repellent should be moved to Section 2.6 (for example, L347 'repelled biting flies'); however, Section 2.6 is entitled "Repellents and Toxicants," and there's emphasis on plant derivatives in Section 2.6. These sections (2.6 and 4.3) should be revisited and clarified.
L359: "...considered economically unfeasible..."? "Too risky" sounds as if it's not going to work.
L376: "...noted that stable fly populations can be suppressed by caring..."
Since most the pubs cited in Section 5.1 are from late 1930s, 1960s, 1970s, it would be helpful to briefly comment that the sanitation efforts are still highly relevant.
L393: define N
L395: "...applied to feedlot..."?
L397: "...have shown reduction in number of..."
Author Response
Title: An Historical Review of Chemical, Biological and Cultural Control Options used against the Stable Fly (Diptera: Muscidae) by David F Cook
Please see the attachment files (3) for responses to each reviewer's comments
General Comments and Changes to the MS
I have changed the paper title from “An Historical Review” to “A Historical Review…”
The most dramatic change I have made to the MS is to now include all information on stable fly trapping systems and methodologies and SIT (Sterile insect Technique) use against stable flies. This makes the 3 sections in the review, i.e., Chemical, Biological and Cultural far more balanced.
Specifically then, under Cultural Control, there is now the following new sections:
5.1. Trapping
5.1.1.Electrocutor Traps
5.1.2. Walk-Through Traps for Livestock
5.1.3. Modification of other Biting Fly Traps
5.1.4. Comparative Trap Studies
5.1.5. Toxic Traps
5.1.6. Effective Stable Fly Control through Trapping
5.2. Sanitation
5.3. Physical Barriers
5.4. Livestock Protection
5.5. Manure Amendments
5.6 Animal Bedding
I also moved the section Integrated Pest Management to after Cultural Control as it involves all 3 of the former sections on Chemical, Biological and Cultural Control and is now stands alone as Section 6.
This now makes the review complete in all respects on any management option used against this pest fly over the past 120 years.

Reviewer 2 Report
The article includes much of the relevant information but in general does not go into too much detail and I think that is a shame. I also think it is a shame to not include the items deliberately excluded, at least in some context.
I think the authors should include some of the new material recently published on zebra stripes:
https://journals.plos.org/plosone/article/file?id=10.1371/journal.pone.0223447&type=printable
In addition, I think you should discuss how a lot of new papers have attempted to include the effect of making changes on behavioral disturbance in the affected animals, see papers below.
https://www.journalofdairyscience.org/article/S0022-0302(16)30647-6/pdf
https://www.journalofdairyscience.org/article/S0022-0302(19)31008-2/pdf
https://www.sciencedirect.com/science/article/pii/S0168159118304040
https://www.sciencedirect.com/science/article/pii/S0737080618301291?via%3Dihub
https://www.sciencedirect.com/science/article/pii/S0304401717303412
With regards to formatting, I think the authors need to reformat the references following the author guidelines and go through the whole manuscript and incorporate references into the text where the reference has been used to replace an author name. For example, you cannot say:
"The use of insecticides against stable flies dates back to the early 1900’s where the first record in the literature on stable fly control was by [6] who wrote about protecting cows from flies."
You should say:
"The use of insecticides against stable flies dates back to the early 1900’s where the first record in the literature on stable fly control was by Beach and Clark [6] who wrote about protecting cows from flies."
I have noted throughout the manuscript where this occurs.
I have made many comments in the attached PDF that I believe should be addressed prior to publication.

Author Response
General Comments
I changed the paper title from “An Historical Review” to “A Historical Review…”
The most dramatic change I have made to the MS is to now include all information on stable fly trapping systems and methodologies and SIT (Sterile insect Technique) use against stable flies. This makes the 3 sections in the review, i.e., Chemical, Biological and Cultural far more balanced.
Specifically then, under Cultural Control, there is now the following new sections:
5.1. Trapping
5.1.1.Electrocutor Traps
5.1.2. Walk-Through Traps for Livestock
5.1.3. Modification of other Biting Fly Traps
5.1.4. Comparative Trap Studies
5.1.5. Toxic Traps
5.1.6. Effective Stable Fly Control through Trapping
5.2. Sanitation
5.3. Physical Barriers
5.4. Livestock Protection
5.5. Manure Amendments
5.6. Animal Bedding
I also moved the section Integrated Pest Management to after Cultural Control as it involves all 3 of the former sections on Chemical, Biological and Cultural Control and is now stands alone as Section 6.
This now makes the review complete in all respects on any management option used against this pest fly over the past 120 years.

Reviewer 3 Report
This paper provides an interesting and in many way, and important contribution to stable fly management, by bringing together in one document a comprehensive survey of stable fly management methods. As such, I do think this work should be published when the writing is corrected. I have provided some changes, but stopped editing because the same errors appear throughout this document. While each individual grammatical error is minor, the aggregate of them makes this a very hard paper to read.
For example, the Abstract needs to be rewritten, it contains lots of grammatical errors, and awkward phrases, such as:
line
11 stable fly should be plural as stable flies
15 human lifestyle should be plural
16 17 options used twice in same sentence
18 use of passive tense is irksome, instead of "This paper presents a comprehensive review of control 17 options from both an historical and technical perspective for managing this pest." write "This paper reviews 17 control options from both an historical and technical perspective for managing this pest."
20 sentence does not make sense "and to substrate"
21 stable flies not stable fly, and collated is wrong word
The Introduction:
31 situations is the wrong word, should be places
35 not all biting flies are stable flies,
37 declared is the wrong word
39 should be stable flies
43 national should not have a capital letter
44 2000 million is 2 billion, this sentence is grammatically wrong, and per-urban is not a word
etc.
I do not think it will be hard to fix, it just needs to be proof read carefully.
Author Response
General Comments:
I changed the paper title from “An Historical Review” to “A Historical Review…”
The most dramatic change I have made to the MS is to now include all information on stable fly trapping systems and methodologies and SIT (Sterile insect Technique) use against stable flies. This makes the 3 sections in the review, i.e., Chemical, Biological and Cultural far more balanced.
Specifically then, under Cultural Control, there is now the following new sections:
5.1. Trapping
5.1.1.Electrocutor Traps
5.1.2. Walk-Through Traps for Livestock
5.1.3. Modification of other Biting Fly Traps
5.1.4. Comparative Trap Studies
5.1.5. Toxic Traps
5.1.6. Effective Stable Fly Control through Trapping
5.2. Sanitation
5.3. Physical Barriers
5.4. Livestock Protection
5.5. Manure Amendments
5.6. Animal Bedding
I also moved the section Integrated Pest Management to after Cultural Control as it involves all 3 of the former sections on Chemical, Biological and Cultural Control and is now stands alone as Section 6.
This now makes the review complete in all respects on any management option used against this pest fly over the past 120 years.
